# Fetal Electrocardiogram Extraction from the Mother’s Abdominal Signal Using the Ensemble Kalman Filter

**DOI:** 10.3390/s22072788

**Published:** 2022-04-05

**Authors:** Sadaf Sarafan, Tai Le, Michael P. H. Lau, Afshan Hameed, Tadesse Ghirmai, Hung Cao

**Affiliations:** 1Department of Electrical Engineering and Computer Science, University of California Irvine, Irvine, CA 92697, USA; ssarafan@uci.edu (S.S.); tail3@uci.edu (T.L.); 2Sensoriis, Inc., Edmonds, WA 98026, USA; mlau@sensoriis.com; 3Obstetrics & Gynecology, Medical Center, University of California Irvine, Irvine, CA 92868, USA; ahameed@hs.uci.edu; 4Division of Engineering and Mathematics, Bothell Campus, University of Washington, Bothell, WA 98026, USA; tadg@uw.edu; 5Department of Biomedical Engineering, University of California Irvine, Irvine, CA 92617, USA

**Keywords:** fetal ecg extraction, fetal monitoring, ensemble kalman filter (EnKF), signal processing

## Abstract

Fetal electrocardiogram (fECG) assessment is essential throughout pregnancy to monitor the wellbeing and development of the fetus, and to possibly diagnose potential congenital heart defects. Due to the high noise incorporated in the abdominal ECG (aECG) signals, the extraction of fECG has been challenging. And it is even a lot more difficult for fECG extraction if only one channel of aECG is provided, i.e., in a compact patch device. In this paper, we propose a novel algorithm based on the Ensemble Kalman filter (EnKF) for non-invasive fECG extraction from a single-channel aECG signal. To assess the performance of the proposed algorithm, we used our own clinical data, obtained from a pilot study with 10 subjects each of 20 min recording, and data from the PhysioNet 2013 Challenge bank with labeled QRS complex annotations. The proposed methodology shows the average positive predictive value (PPV) of 97.59%, sensitivity (SE) of 96.91%, and F1-score of 97.25% from the PhysioNet 2013 Challenge bank. Our results also indicate that the proposed algorithm is reliable and effective, and it outperforms the recently proposed extended Kalman filter (EKF) based algorithm.

## 1. Introduction

A national study reported by the Centers for Disease Control (CDC) showed that the U.S. fetal mortality rate remained unchanged from 2006 through 2013 from the rate of 6.05 to 5.96 per 1000 births [1]. Thus, fetal monitoring is essential throughout pregnancy for recognition of elements that might imperil the life of the fetus and mother. However, the 2015 Cochrane review of gold standard antenatal cardiotocography (CTG) for fetal assessment showed no clear evidence that it improves perinatal outcome [2].

The current global COVID-19 pandemic has led to critical demand for person-centered healthcare instead of hospital-centered healthcare system. Expectant moms usually come to the obstetric clinics many times during pregnancy for checkups. According to the CDC, a study on 461,825 women with COVID-19 showed that pregnant women with COVID-19 are more likely admitted to an intensive care unit, receiving invasive ventilation, extracorporeal membrane oxygenation, or die compared with non-pregnant women [3]. This would stop pregnant women from having physiological measurements, ultrasound examination, or non-stress test, which are critical for both the mom and the unborn baby. This calls for an urgent need for novel home-based tools and systems for reliable prenatal monitoring.

There are already existing ultrasound-based consumer products for fHR assessment in the home setting. Those require active scanning over the abdomen coupling with ultrasound gel to locate the fetal heart to obtain the fHR, which is highly technique dependent, easily make difficult by maternal of fetal movement and larger body size. Thus, it is especially challenging for non-medical persons to administer it. Further, the U.S. Food and Drug Administration (FDA) issued a warning in 2014 against the use such ultrasound fHR home monitors [4], citing safety concerns over their repeated use.

The fetal electrocardiogram (fECG) can also be used to assess fetal wellbeing and even enable further diagnoses. Precisely-extracted fECG will provide vital information about fHR, fetal development, fetal maturity, and the existence of abnormalities or distress during pregnancy [5]. However, continuous and noninvasive fECG monitoring has remained challenging in the research community in terms of both signal acquisition and processing. There are several methods proposed for extracting fECG from aECG. These methods can be generally classified into three groups: blind source separation (BSS), template subtraction, and filtering techniques. The BSS methods include methods such as parallel linear predictor (PLP) filter, principal component analysis (PCA), independent component analysis (ICA), and periodic component analysis (πCA) [6,7,8,9]. The BSS methods consider that the abdominal signal is a combination of fECG, mECG, and noises [10]. Although these methods perform greatly for fECG extraction, they need multiple-channel aECG signals, which makes them unsuitable for continuous non-invasive fECG monitoring. In addition, after extraction, the order of the separated independent component could not be determined. Thus, it is challenging to identify the fECG component for further processing [11]. Therefore, the BSS methods usually require the determination of other parameters (e.g., *t*-test, correlation coefficient, heart rate) to automatically identify the extracted components [12,13,14]. Template subtraction (TS) is another widely used approach. The method involves subtracting a synthetic mECG generated by estimating the QRS complex waveform (mQRS) of mECG from the abdomen signal [15,16,17,18,19,20]. The main challenge of this method involves mQRS detection [21], which becomes more challenging if the fetal R waves overlap with the maternal R waves. This drawback degrades the effectiveness of the template subtraction method for fECG extraction. The popular filtering techniques include adaptive filtering [22,23,24,25], Kalman filtering [26,27,28], and wavelet transform [29,30]. These filtering techniques are mostly and effectively applied for denoising of single-channel ECG signals. Adaptive filtering-based algorithms have been proposed for fECG extraction [31]. Such methods, however, require additional reference signals for separating the different components of the aECG.

Our group has been developing abdominal patch devices and systems for the acquisition and extraction of fECG [12,31,32]. As we aim to make the device compact and unobtrusive, a single recording channel is desired for saving space; consequently, fECG extraction is almost impossible, especially with the presence of motion noise [33]. In this case, there are multiple signals buried in the single-channel aECG including maternal ECG (mECG), fECG, maternal muscle activity, fetal movement activity, and noise. Here, we propose a novel algorithm based on the Ensemble Kalman filter (EnKF) to extract fECG from a single-channel aECG signal. The EnKF is an approximate filtering method that is used for state estimation of large-scale nonlinear systems. Specifically, the EnKF is a Sequential Monte Carlo (SMC) method, and it shows a better performance, especially in systems with strong nonlinearity, than the popular Extended Kalman filter (EKF) that applies analytic linearization. We studied and compared the performance of EKF and EnKF using our aECG data obtained from 10 pregnant subjects, and data from the PhysioNet database. The results demonstrated that our EnKF algorithm is more effective and outperforms the regular EKF in accurately extracting fetal ECG.

## 2. Theory and Methods

### 2.1. Ensemble Kalman Filter (EnKF)

EnKF is a variant of the celebrated Kalman filter used to estimate time-varying parameters in problems that arise in various disciplines [34]. It is applicable for problems that can be represented as dynamic systems and formulated in a state-space model with unknown time-varying state parameters. When the state-space model of the system is linear and Gaussian, the optimal estimate of the state parameters can be obtained using the Kalman filter [35]. However, when the problem is nonlinear or non-Gaussian, other variants of the Kalman filter, such as the EnKF, are used to obtain close-to-optimal solutions [36]. Our EnKF algorithm is developed by considering a Bayesian filtering framework and formulating the fECG extraction problem as a dynamic system whose state and measurement equations are represented in a state-space form. The dynamic model is adopted from the models proposed by McSharry et al. [37] and later discretized by Sameni [38].

Suppose the unknown time-varying state vector of a dynamic state-space model is denoted by $x_{n}\in R^{D_{x}}$ where n=1,2,…,N represents time instants and $D_{x}\;$ represents the dimension of $x_{n}$. We assume that $x_{n}$ has a Markovian property, and its evolution is given by:(1)$$x_{n}=f_{n}\left(x_{n-1}\right)+u_{n},$$
where $f\left(.\right)$ represents a state function which, in general, is nonlinear, and $u_{n}$ denotes the state noise vector with a known probability density function (pdf). Furthermore, the observation equation of the state-space model is given by:(2)$$y_{n}=h_{n}\left(x_{n}\right)+w_{n},$$
where $y_{n}\in R^{D_{y}}$ denotes the measurement vector obtained at time n, $D_{y}$ represents the dimension of the vector $y_{n}$, and $w_{n}$ denotes the measurement noise vector whose pdf is assumed known.

Given the state-space model (1) and (2), our objective is to make a sequential estimate of the evolution of the state vector $x_{1:n}=\{x_{1},\ldots ,x_{n}$} in real-time as the measurement vector denoted by $y_{1:n}=\{y_{1},\ldots ,y_{n}$} becomes available. 

The Ensemble Kalman filter is a variant of the Kalman filter where the state error statistics are approximated using the Monte Carlo method. Recall that if the state Equation (1) and the measurement Equation (2) are linear, and the state noise $u_{n}$ and measurement noise $w_{n}$ are Gaussian, the optimal estimate of the state vector can be analytically obtained using Kalman filter. To understand the EnKF, let us review the Kalman filter algorithm’s two steps: the time update and the measurement update. For convenience, we rewrite the state and measurement equations for a linear and Gaussian systems as follows:(3)$$\left\{\begin{array}{c} y_{n}=Hx_{n}+w_{n}\\ x_{n}=Fx_{n-1}+u_{n} \end{array}\right.,$$
where F is a $D_{x}\times D_{x}$ matrix, H is a $D_{y}\times D_{x}$ matrix, and $u_{n}$ and $w_{n}\;$ are zero-mean Gaussian probability densities with covariances $Q_{u}\;$ and $Q_{w}$, respectively.

The time update step of the Kalman filter algorithm obtains the predicted value of the state vector and the covariance of its error. The predicted state vector and the covariance of its error at time n are computed from propagating their corresponding values at time n−1 through the state dynamics as follows:(4)$${\hat{x}}_{n|n-1}=F{\hat{x}}_{n-1}\;P_{n|n-1}=FP_{n-1}F^{T}+Q_{u}$$
where ${\hat{x}}_{n|n-1}\;$ is the predicted value of the state vector, $P_{n|n-1}$ is the covariance of the error of the predicted state vector given by $P_{n|n-1}=E\left[\left({\hat{x}}_{n}-{\hat{x}}_{n|n-1}\right){\left({\hat{x}}_{n}-{\hat{x}}_{n|n-1}\right)}^{T}\right]$, and $E\left[.\right]$ is the expectation operator. Once the measurement vector is received, the measurement update step is used to compute the estimate of the state vector, ${\hat{x}}_{n}$ and the covariance of the error of the state vector, $P_{n}$, by applying corrections to the corresponding predicted values based on the measurement obtained as follows:(5)$$K_{n}=P_{n|n-1}H^{T}{\left(HP_{n|n-1}H^{T}+Q_{w}\right)}^{-1}\;{\hat{x}}_{n}={\hat{x}}_{n|n-1}+K_{n}\left(y_{n}-H{\hat{x}}_{n|n-1}\right)\;P_{n}=\left(I-K_{n}H\right)P_{n|n-1}$$
where $K_{n}\;$ is the Kalman gain, and I is a diagonal matrix. We note that the Kalman gain can be expressed as
(6)$$K_{n}=P_{xy,n}P_{yy,n}{}^{-1},$$
where $P_{xy,n}\;$ is the cross-covariance of the error of $x_{n|n-1}\;$ and $y_{n}$, and $P_{yy,n}\;$ is the covariance of the error of $y_{n}$. These covariances are given by:(7)$$P_{xy,n}=E\left[\left({\hat{x}}_{n}-{\hat{x}}_{n|n-1}\right){\left(y_{n}-y_{n|n-1}\right)}^{T}\right]\;P_{yy,n}=E\left[\left(y_{n}-y_{n|n-1}\right){\left(y_{n}-y_{n|n-1}\right)}^{T}\right]$$

When the state equation and measurement equation are not linear, the above equations of the covariances of the errors, $P_{n|n-1}\;$ and $P_{n}$, as well as the equation of the Kalman gain, $K_{n}$, are not valid, and, therefore, generally, the Kalman gain cannot be analytically determined. A common approach of circumventing such problems in nonlinear state-space models is to apply extended Kalman filter (EKF), which is a variant of the Kalman filter, that approximates the state and measurement equations by linearizing them using the Taylor series. Consequently, the EKF approximates the posterior probability density of the state vector by a Gaussian distribution. When the true posterior distribution of the state vector is not ‘close’ to Guassian, such approximations may not be valid and the EKF may diverge. In such cases, sequential Monte Carlo methods show superior performance over EKF [39]. 

EnKF is the Monte Carlo-based Kalman filter which can be used for nonlinear and non-Gaussian models. The underlying idea of the method is to approximate the Kalman gain and state vector propagations using Monte Carlo technique. EnKF computes the Kalman gain by approximating $P_{xy,n}\;$ and $P_{yy,n}\;$ using their corresponding sample covariances, ${\hat{P}}_{xy,n}$ and ${\hat{P}}_{yy,n}$. To do so, *N* number of ensembles, $\left\{x^{\left(i\right)}{}_{n|n-1}\right\}\begin{array}{c} N\\ i=1 \end{array}$, are first drawn from the prior probability density of the state vector, $p\left(x_{n|n-1}\right)$, which has the same probability distribution function as the state noise with a mean of $f\left(x^{\left(i\right)}{}_{n-1}\right)$. Once the ensembles are generated, the sample covariances of the errors are computed as follows:(8)$${\hat{P}}_{xy,n}=\frac{1}{N}\overset{N}{\underset{i}{\sum }}\left(x^{\left(i\right)}{}_{n|n-1}-{\underline{x}}_{n}\right){\left(y^{\left(i\right)}{}_{n|n-1}-{\underline{y}}_{n}\right)}^{T}\;{\hat{P}}_{yy,n}=\frac{1}{N}\overset{N}{\underset{i}{\sum }}\left(y^{\left(i\right)}{}_{n|n-1}-{\underline{y}}_{n}\right){\left(y^{\left(i\right)}{}_{n|n-1}-{\underline{y}}_{n}\right)}^{T}$$
where $y^{\left(i\right)}{}_{n|n-1}=h\left(x^{\left(i\right)}{}_{n|n-1}\right)$, ${\underline{x}}_{n}=\frac{1}{N}\overset{N}{\underset{i}{\sum }}x^{\left(i\right)}{}_{n|n-1}$
(9)$${\underline{y}}_{n}=\frac{1}{N}\sum_{i}^{N}y^{\left(i\right)}{}_{n|n-1},$$

Then, the Kalman gain is approximated by:(10)$${\hat{K}}_{n}={\hat{P}}_{xy,n}{\left({\hat{P}}_{yy,n}\right)}^{-1},$$
and the ensembles of the state vector, {$x^{\left(i\right)}{}_{n|n-1}$}, are computed as
(11)$$x^{\left(i\right)}{}_{n}=x^{\left(i\right)}{}_{n|n-1}+{\hat{K}}_{n}\left(y_{n}+v^{\left(i\right)}{}_{n}-y^{\left(i\right)}{}_{n|n-1}\right),$$
where $v^{\left(i\right)}{}_{n}$ are samples obtained from Gaussian distribution with mean ${\underline{y}}_{n}$ and covariance $Q_{w}$.

Once the ensembles of the state vector are computed, the estimate of the state vector is obtained by taking the averages of ensembles as follows:(12)$${\hat{x}}_{n}=\frac{1}{N}\sum_{i}^{N}\left(x^{\left(i\right)}{}_{n}\right),$$

### 2.2. State-Space Model of a Synthetic ECG

In the work reported in, McSharry et al. proposed a dynamic model which consists of a set of nonlinear state equations to generate synthetic ECG signals in the Cartesian coordinate system [37]. Further, Sameni et al. transformed the model to a polar coordinate system and provided a convenient discrete-time mathematical model [38]. The model represents an ECG signal by a sum of five Gaussian functions, each corresponding to the five waves of an ECG signal, namely P, Q, R, S, T waves. The state vector of the dynamic model is defined by $x_{k}={\left[\theta_{k},z_{k}\right]}^{T}$, and the state equation is given by:(13)$$\left\{\begin{array}{c} \theta_{k}=\left(\theta_{k-1}+\omega .\Delta \right)mod\;2\pi \\ z_{k}=-\sum_{i\in \left[P,\;Q,\;R,\;S,\;T\right]}\frac{\alpha_{i}\Delta \theta_{i}\omega .\Delta }{b_{i}{}^{2}}\exp\left(-\frac{\Delta \theta^{2}{}_{i}}{2b_{i}{}^{2}}\right)+z_{k-1}+\eta_{k} \end{array}\right.$$
where $\Delta \theta_{i}=\left(\theta_{k}-\theta_{i}\right)mod\;2\pi$ is the phase increment, Δ is the sampling period, $\eta_{k}$ is the state noise, ω is the angular velocity of the trajectory as it moves around the limit cycle, and $\alpha_{i}$, $b_{i}$ and $\theta_{i}$ represent the amplitude, width, and center of the Gaussian functions of the five PQRST waves, respectively.

The measurement vector is defined by $y_{k}={\left[\phi_{k},s_{k}\right]}^{T}$, where $\phi_{k}$ is the observed phase representing the linear time wrapping of the R-R time interval into $\left[0,\;2\pi \right]$, and $s_{k}$ is the observed amplitude. The measurement equation is given by
(14)$$\left\{\begin{array}{c} \phi_{k}=\theta_{k}+u_{k}\\ s_{k}=z_{k}+v_{k} \end{array}\right.$$
where $u_{k}$ and $v_{k}$ denote the measurement noises.

### 2.3. EnKF Based fECG Extraction Algorithm

Given the state-space model (13) and (14), the EnKF algorithm in Figure 1 is applied to filter out the mECG from aECG, assuming the remaining signal composed of fECG and noise is Gaussian distributed. The extracted mECG signal is then subtracted from aECG to obtain a noisy fECG signal. Finally, the EnKF algorithm is applied to the residual signal to denoise the fECG signal. Before running the extraction algorithm, the acquired aECG signals are processed to remove the baseline wander, the powerline, and ambient interferences. The baseline wander is removed using a lowpass filter, and a notch filter is used to suppress the powerline interference noise. We also applied the wavelet filtering and thresholding technique and compare [40]. This will be discussed in detail in later sections. Finally, the fetal QRS complex (fQRS) is detected using the Pan-Tompkin algorithm [32,38].

### 2.4. Data for Testing

Three different datasets were used to test the performance of the proposed algorithm and to compare with the EKF. They are the data obtained from the PhysioNet 2013 Challenge database, the same data with motion noise added, and our own data obtained in a recent pilot study with pregnant subjects.

#### 2.4.1. Challenge Databank

The PhysioNet 2013 Challenge databank used in this work consists of 75 recordings, excluding a number of recordings (a33, a38, a47, a52, a71, and a74) that had inaccurate reference annotations. Each recording comprises four different abdominal signals. All signals have been sampled at 1 kHz and recorded for 60 s. In each case, reference annotations marking the locations of each fetal QRS complex were produced, usually with reference to a direct fECG signal, acquired from a fetal scalp electrode. The reference annotations are produced by a team of experts manually [41].

#### 2.4.2. Modified Signals with Motional Artifacts Added

The aECG recordings acquired in real-life settings would possess a variety of interferences, including motion artifacts. The online databank, however, was obtained in the clinical setting, where motion noise was mostly non-existent since the subjects were in a resting position. We added realistic motion artifacts to the online databank. The realistic motion noise generation process is described in our previous paper [32], to mimic real-life scenarios. In this experiment, the ECG data were recorded from a healthy subject during different types of activities. Then, the normalization had been used to reinsure aECG and the motion noise have realistic amplitudes. The recorded data were normalized between −1 and 1. Then, the motion noise was extracted by using the EKF. Before adding the motion noise to the new aECG, the aECG data should be normalized with the same threshold. In the last step, the generated motion noises are added to the PhysioNet 2013 Challenge databank. Figure 2 illustrates the modified data with motion noise process.

#### 2.4.3. Our Human Data

We developed the gen-2 ‘fetal monitoring patch’ containing non-contact electrodes (NCEs), electronics, and secure communication with a smart device via Bluetooth Low Energy (BLE) [12]. The compact patch is ~4 inches long and unobtrusive, thus, it can be integrated on or inside maternity garment. The gen-2 patch has one single NCE channel to collect the aECG of the pregnant subject. The collected data are sent to an Android app connected to a cloud server for analytics. The system was validated on 10 pregnant women between 28 and 34 weeks of gestation in the UCI Medical Center. Each subject was resting on a chair during recording, and a maternity belt with the fECG patch attached is worn so that the NCEs are located on the abdominal area below the navel. The data were then collected in 5 min for each posture. More details are described in our previous paper [42]. All clinical recordings were collected anonymously under Institutional Review Board (IRB) approval #2020-6342 at the University of California, Irvine (UCI).

### 2.5. Comparison Criteria

The performance of the extraction methods is assessed by comparing the beat-to-beat length of the extracted fECG QRS complex and the corresponding annotated data. According to the American National Standards Institute/Association for the Advancement of Medical Instrumentation (ANSI/AAMI) guideline, sensitivity (SE), positive predictive value (PPV), and the accuracy measure (F1 score) which is the harmonic mean of PPV and SE, were used for assessment. These statistical indices are computed as follows
(15)$$\mathrm{SE}=\frac{TP}{TP+FN}\;\mathrm{PPV}=\frac{TP}{TP+FP}\;F1=\frac{2.TP}{2.TP+FN+FP}$$
where *TP*, *FP*, and *FN* are the number of true positive, false positive, and false negative, respectively.

## 3. Results

The execution time of each algorithm is calculated from the start of the pre-processing to the end of the fECG R-peak detection. The EnKF with ensemble size of 70 has the same execution time such as the EKF. Hence, the proposed EnKF-based algorithm was run with an ensemble size of N = 70 for single-channel signals from the aforementioned datasets. For comparison purposes, the EKF-based algorithm using the same synthetic ECG parameters and datasets was also carried out. The three-dimensional trajectory generated from (13) consists of a unit-radius circular limit cycle that goes up and down when it approaches one of the P, Q, R, S, or T points (Figure 3). The projection of these trajectory points on the z-axis gives a synthetic ECG signal. Figure 3 shows plots of the ECG signals versus the assigned phases in polar coordinates on the unit-radius circle. The figure depicts a typical phase-wrapped aECG signal (a), EKF extracted mECG signal (b), and EnKF extracted mECG signal (c) plotted using a sample signal taken from the PhysioNet database.

Examples of two abdominal ECG signals of records “a01” and “a03”, and fetal signals extracted using the EKF and EnKF, are shown in Figure 4. Panels a and d are the original signals, while b–e and c–f are fQRS extracted using the EKF and EnKF approaches, respectively. The fQRS annotation is shown in an orange asterisk (*). The red arrows show the places that fetal QRS was wrongly detected. The blue arrows show the missing fetal QRS. It can be seen that the detected fQRS (fetal R-peaks) follow the annotated ones with high accuracy. It is worth noting that in “a01” (Figure 4d), the fetal QRS complexes are reversed due to electrode placement, but it did not affect the extraction algorithms. It should be emphasized that in the case of overlapping of the fetal QRS and maternal QRS, the EnKF algorithm still gives favorable results. Comparison between Figure 4b,c show that the EKF failed when maternal and fetal QRS complexes overlap in time (e.g., at t ≅ 21.8 s and t ≅ 23.7 s) for ‘a03’, while our EnKF method still performed successfully. It can be also seen that the EnKF functions reasonably well with the presence of noise ((e.g., at t ≅ 19.7 s for ‘a01’, Figure 4d–f).

Figure 5 depicts the fECG extraction results from the modified PhysioNet data with added motion artifacts. As seen, the fECG extracted by the EKF was incorrect and its F1 score was reduced significantly (average F1 = 78, see Table 1). fQRS complexes extracted by the EnKF, however, are still visible in most time points, yielding a favorable F1 score of 89.

Figure 6 shows the performance of the proposed algorithm, the EnKF, and the EKF on one sample from our clinical data. Panels a–c show the original and extracted signals, while panels d–f are signals with extensive preprocessing. Specifically, in Figure 6d, a lowpass filter (with a cut-off frequency of 1 Hz), a notch filter, and a Wavelet filter (a 10 level 1-D stationary wavelet decomposition with Coiflet mother wavelet) were used to suppress the background noise and artifacts. The signal extracted from the preprocessed data with the EKF and EnKF algorithms are shown in Figure 6e,f. In Figure 6f, we also further notice that fetal characteristic waves, such as P and T waves, may be conserved. This judgment is also strengthened by the EnKF extraction carried out on the original PhysioNet database in Figure 4f, where the conserved features are probably fECG waves. Table 1 presents the average F1, average PPV, and average SE results in our own clinical aECG records. The average F1, PPV, and SE indices of our proposed EnKF are 94.3%, 100%, and 89.2%; while those using the EKF method are 82.3%, 71.4%, and 100%, respectively.

Table 1 shows the average F1 scores, the average PPE, and the average SV of the performance of the EnKF and EKF algorithms. These statistical indices are computed by determining the accuracy, TP, FP and FN, of the locations of the R-peaks obtained by the EKF and EnKF algorithms against the reference annotations. The F1 score, PPV, and SE are computed using 68 one-minute aECG records from PhysioNet 2013 Challenge databank and our own clinical data. The results shown in Table 1 indicate that the EnKF method is reliable on its own. In all cases, the EnKF outperforms the EKF.

Another parameter that should be taken into account is the computational complexity. The computational complexity of the EnKF algorithm is proportional to the number of ensembles used. In our simulations, we have observed that increasing the size of the ensemble by more than 70 does not improve the performance of the algorithm significantly. The F1 score obtained when the algorithm was run for ensemble sizes between 5 to 350 always remained in the range between 94.5% and 98.6% for all of the 68 aECG records obtained from the PhysioNet database.

## 4. Discussion and Conclusions

The current fECG extraction methods from a single-channel signal are not robust when particularly (i) the fECG and mECG waveforms temporally overlap, and (ii) the amplitude of fECG is low compared to the noise level. Recently, an EKF-based algorithm was proposed for the extraction fECG from a single-channel aECG signal [43]. Generally, those EKF-based algorithms are less effective as at every instant of time, they approximate the posterior probability density of the parameter of interest by a Gaussian distribution. When the true posterior density is not Gaussian, sequential Monte Carlo (SMC) filtering methods show superior performance over EKF methods. Therefore, here, our proposed EnKF is an SMC-based method, for noninvasive extraction of fECG from the single-channel aECG. As described above, our EnKF-based algorithm exhibited robust performance when tested using public online data as well as our own clinical data. Ten records, including 20-min aECG from our own clinical database, 68 records, including 1-min aECGs in PhysioNet Challenge 2013, and the online data bank added realistic motion artifacts were used to assess the performance of the proposed method.

As can be seen in Figure 3, the EnKF-based algorithm follows the dynamics of ECG and thus suppresses the noise better than EKF. In the EKF method, since some of the fECG peaks were still incorporated in the estimated mECG (3b), the subtraction between the aECG and mECG will not produce the correct fECG. In contrast, the mECG estimated by EnKF is significantly clearer (3c), which explains the better performance in fECG extraction (Figure 4, Figure 5 and Figure 6 and Table 1).

Our experiments proved that the EnKF-based algorithm is a robust method and has superior performance over the EKF for extracting fECG in various scenarios (Figure 4, Figure 5 and Figure 6). Results support the expectations that the EnKF not only extracts the fetal signal when the fECG and mECG waveforms temporally overlap, but it also potentially extracts accurate fECG signal with characteristic waves such as P and T waves. Currently, there is no approach available to extract full-feature fECG, especially with aECG in daily life. Hence, our solution may hold the potential to revolutionize fetal monitoring as it can be used to diagnose potential congenital effects. Currently, this is carried out with genetic test and echocardiogram [44,45,46]. However, those need to be carried out in the clinics and they cannot provide continuous information of the fetal heart over a long period of time. For example, with our fetal ECG patch and this EnKF method, full-feature fECG can be obtained 24/7 in the home setting in the second and third trimesters of pregnancy, which provides information about not only fetal wellbeing and development but also any fetal cardiac anomalies.

Due to the nature of aECG acquisition, the signal is always accompanied by other bioelectric potentials, such as maternal muscle activity, fetal movement activity, and noise [33]. The results shown in Figure 4, Figure 5 and Figure 6 suggested that, prior to fECG extraction, preprocessing is a critical step which needs to be optimized. Specifically, preprocessing may help extraction significantly as it removes noise components, but it may also eliminate low-amplitude precious components such as P and T waves. In Figure 6d, we can see that, when the aECG signal is somewhat over-processed, both EKF and EnKF perform reasonably well; the signal, however, may have lost its intrinsic peaks. The filtering scheme used here is a low-pass filter, a digital Notch filter, and the wavelet filtering and thresholding technique described in [40]. As P and T waves have similar frequency and amplitude with noise components, it is almost impossible to set up optimal thresholds to eliminate the noise and keep the desired waves in all cases. Therefore, we usually carry out this manually until the best performance is obtained. The challenge is the noise level varies from person to person, setting to setting, and even time to time within one person in one setting. To address this, we are planning to add an accelerometer to collect the motion noise as carried out in [12], and explore further algorithms to extract uterine contraction and fetal movement.

The fECG would provide valuable information that could help deliver better prenatal care as well as assisting clinicians in making appropriate and timely decisions during labor. In this context, our EnKF presented in this work may bring significant impacts as a robust and efficient algorithm deployed in compact wearables for fECG assessment in the home setting. Future work includes assessing the fECG patch and the EnKF algorithm in out-of-clinic settings. We are planning to first validate the performance with pregnant subjects during sleep overnight in their home first and then move forward to real-life trials for several days. We also plan to correlate with other physiological signals using machine learning and cloud computing to link the mother’s activity with fetal development, such as the mother’s sleep quality or diet [42,47]. In the future work, the fetal ECG monitoring in twins and multi-fetal prenatal should also be studied.

## Figures and Tables

**Figure 1 sensors-22-02788-f001:**
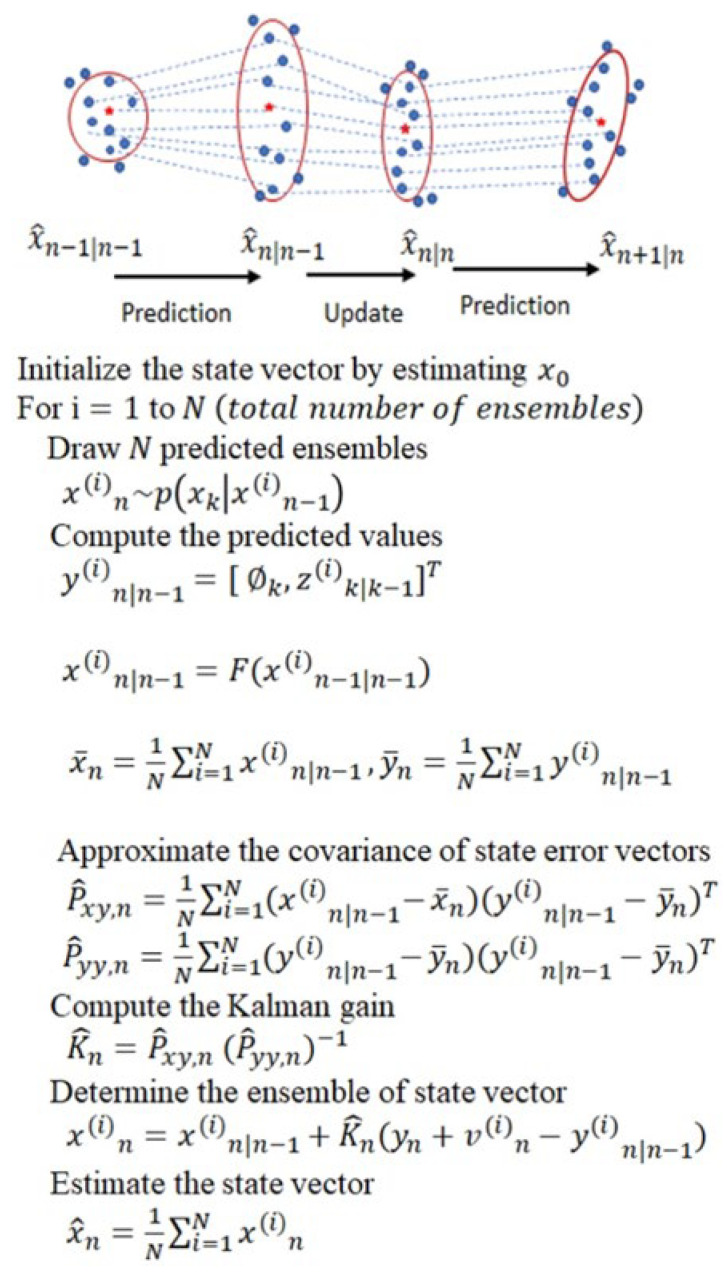
An overview of EnKF. EnKF maintains an ensemble of sample points for the state vector $x_{n}$. It propagates and updates the ensemble to track the distribution of $x_{n}$. The state estimation is conducted by calculating the sample mean (red five-pointed-star) and covariance (red ellipse) of the ensemble.

**Figure 2 sensors-22-02788-f002:**
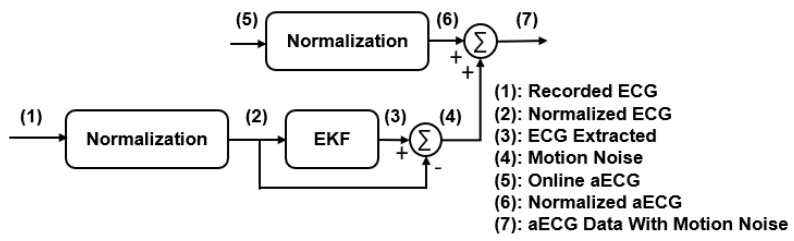
Modified data with motion noise process: (**1**) Recorded ECG; (**2**) normalized ECG between −1 and 1; (**3**) EKF applied for ECG extraction; (**4**) motion noise extracted by subtracting filtered ECG signal from recorded ECG; (**5**) online aECG; (**6**) normalized aECG between −1 and 1; (**7**) the generated motion noises are added to the online aECG data.

**Figure 3 sensors-22-02788-f003:**
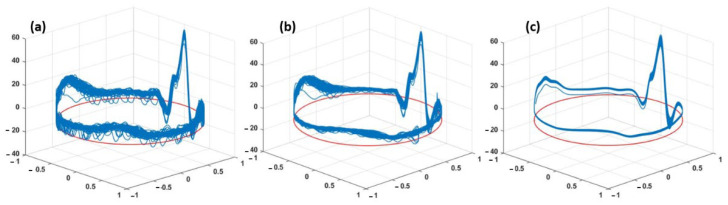
A phase-wrapped ECG signal of records “a01”. (**a**) Abdominal ECG (raw data); (**b**) mECG extracted using EKF; and (**c**) mECG extracted using our EnKF.

**Figure 4 sensors-22-02788-f004:**
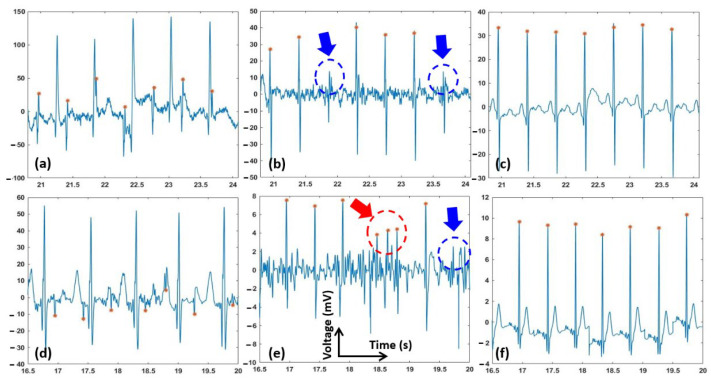
fECG extraction using the EKF and EnKF with the PhysioNet data. The fQRS annotation is shown in orange asterisk (*). The red arrows show the places that fetal QRS was wrongly detected. The blue arrows show the missing fetal QRS. (**a**) Abdominal ECG (raw data) of record “a03”; (**b**) fECG extracted using EKF of record “a03”; (**c**) fECG extracted using EnKF of record “a03”; (**d**) Abdominal ECG (raw data) of record “a01” with reversed fetal QRS complexes; (**e**) fECG extracted using EKF of record “a01”; (**f**) fECG extracted using EnKF of record “a01”.

**Figure 5 sensors-22-02788-f005:**
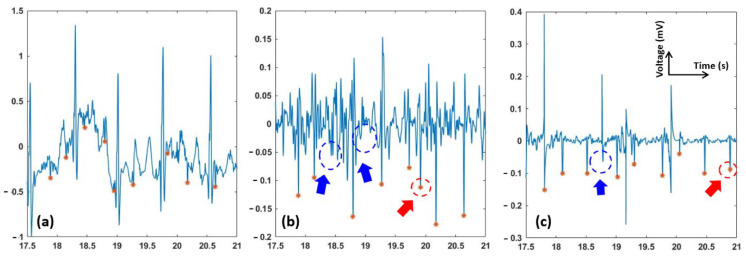
fECG extraction using the EKF and EnKF with the motion artifacts added PhysioNet data. The fQRS annotation is shown in orange asterisk (*). The red arrows show the places that fetal QRS was wrongly detected. The blue arrows show the missing fetal QRS. (**a**) Abdominal ECG (raw data); (**b**) fECG extracted using EKF; and (**c**) fECG extracted using EnKF.

**Figure 6 sensors-22-02788-f006:**
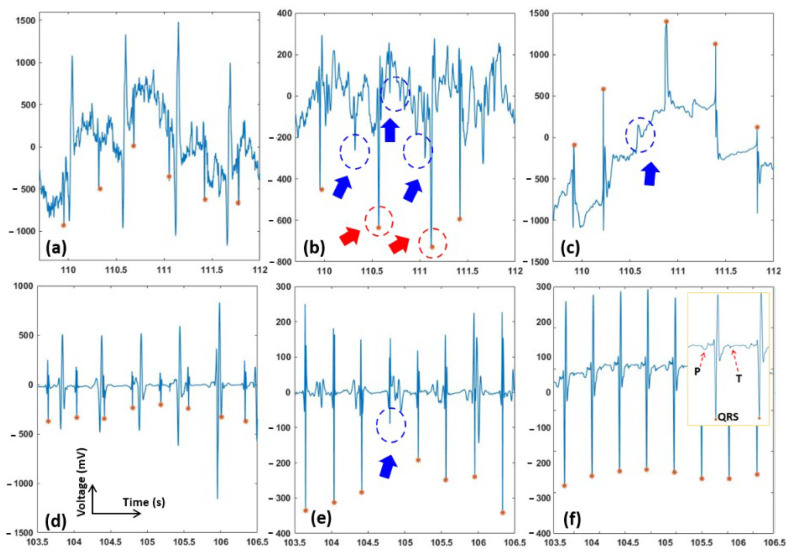
fECG extraction using the EKF and EnKF with our own clinical data. The fQRS annotation is shown in orange asterisk (*). The red arrows show the places that fetal QRS was wrongly detected. The blue arrows show the missing fetal QRS. (**a**) Abdominal ECG (raw data), (**b**) fECG extracted from EKF and (**c**) fECG extracted from EnKF”. (**d**) Abdominal ECG after Wavelet preprocessing; (**e**) fECG extracted using EKF of preprocessed data; and (**f**) fECG extracted using EnKF of preprocessed data. The inset shows additional peaks, likely P and T waves; they were conserved after EnKF extraction.

**Table 1 sensors-22-02788-t001:** Performance of the EKF and EnKF algorithms.

		Average F1 (%)	Average SE (%)	Average PPE (%)
Online Data without Motion Noise	EKF	88.90 ± 5	86.73 ± 5.5	91.16 ± 4.6
	EnKF	97.25 ± 2.4	96.91 ± 0.5	97.59 ± 3.8
Online Data with Motion Noise	EKF	78 ± 6.58	75.38 ± 7.4	80.80 ± 5.1
	EnKF	89.04 ± 3	88.2 ± 1.7	89.9 ± 4.5
Our Clinical Data	EKF	82.3 ± 5.5	100 ± 0.1	71.4 ± 6.4
	EnKF	94.3 ± 1.2	89.2 ± 1.5	100 ± 0.2

## Data Availability

Not applicable.

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
