# Peer review of "Fetal Electrocardiogram Extraction from the Mother’s Abdominal Signal Using the Ensemble Kalman Filter"

_sensors, 2022, doi:10.3390/s22072788_

Round 1

Reviewer 1 Report

The paper “Fetal Electrocardiogram Extraction from the Mother’s Abdominal Signal Using the Ensemble Kalman Filter” describes an Ensemble Kalman Filter-based algorithm (EnKF) for fetal electrocardiogram (fECG) extraction. They examine the algorithm by using public online and clinical data, and compare its performance with Extended Kalman Filter algorithm. There are several points that are better to be improved in order to make the manuscript acceptable for publication. My comments are listed below:

#1. One of the innovations of this work is the EnKF for fECG. Please compare the algorithm with the reference [Sensors & Transducers. 2017;209(2):90-96.] using the similar method in detail. And explain the highlights of this research.

#2. The authors use their custom-built device to collect ECG signals in this work. To verify the validity of raw signals and EnKF processed data, please include the data collected by commercial ECG devices for reference.

#3. For better readability, please include the preprocessing details of each data set, because the preprocessing condition affects the results significantly.

#4. Please include a table that compares the performance of EnKF among other filtering techniques (like the techniques in line 74).

#5. To assess the repeatability of this algorithm, please include standard deviations of the performance, and discuss them.

Author Response

Dear Honorable Reviewer,

Thank you so much for providing invaluable comments and suggestions for our submission. We have been able to incorporate changes to reflect most of the suggestions provided by the reviewer. We have highlighted the changes within the manuscript. Please see the attachment.

Reviewer 2 Report

To be honest, I am quite interested in this study because the fetal heart rate variability and fetal electrocardiogram are considered the most important sources of information about fetal wellbeing and also the authors have proposed an approach for fetal electrocardiogram extraction from the mother’s abdominal signal using the ensemble kalman filter. The manuscript is well written and organized. In addition, the results of the manuscript are well presented.

However, to improve the manuscript, the authors should incorporate the following comments:

  1. It will be interesting and helpful to find in the introductory sections some quantitative comparison between several methods available to extract fECG from raw ECG values.
  2. Can authors provide a Block diagram / Process Diagram for the fECG recording, extraction and comparison procedure they used?
  3. Can authors show graphical results for the removal of the baseline wander, the power-line, and ambient interferences?
  4. Please cite PhysioNet 2013 Challenge database you referred in reference section.
  5. How authors fix the normal ranges and values of the fetal electrocardiogram parameters of 18 to 24 weeks of gestation?
  6. Do authors have idea of extracting fECG from a pregnancy women having more than one child?
  7. Please include some most recent references in your manuscript which are already available online.
  8. In line number 332, after the word extraction, of is missing. Please add.

Author Response

(The authors gave the same response as above.)

Reviewer 3 Report

I consider that the idea of this study, to find a novel algorithm based on the Ensemble Kalman Filter (EnKF) for non-invasive fECG extraction from a single-channel aECG signal is very interesting and with important clinical practice, considering that is essential throughout pregnancy to monitor the wellbeing and development of the fetus and to possibly diagnose potential congenital heart defects. The article is well-written and comprehensive. The results are clearly and transparently presented and the set goals correspond to the conclusions. I consider that the findings are interesting and that the results obtained can make significant contributions to further large studies. I consider that the article may be accepted for publication in its current form.

Author Response

Dear Honorable Reviewer,

Thank you so much for providing invaluable comments.

Reviewer 4 Report

The manuscript is devoted to the fetal electrocardiogram extraction using ensemble Kalman filter. The task is actual. The aim of the research is to develop and to examine the new method for the extraction of fetal ECG from a single-channel mother’s abdominal ECG signal.

The paper consists of theoretical part and experimental part. Theoretical part consists of the description of the idea of Ensemble Kalman Filter application. The experimental part includes experimental investigation of the proposed method. The authors compared the results of fetal ECG extraction using Extended Kalman Filter and Ensemble Kalman Filter on three datasets (data from the PhysioNet database, modified data from the PhysioNet and original aECG data obtained by the authors). Comparison with only one existing method is a limitation of the experimental research. The idea and the experimental results are clear. The references are relevant and up-to-date but more than half of them are older than 5 years.

This paper will surely be interesting for researchers and engineers in the field of fetal ECG monitoring systems development.

The subject of the article can be ascribed to the subjects of the journal Sensors, specifically the Signal processing, data fusion and deep learning in sensor systems.

Some remarks to the authors:

  • The sentence “However, when the problem is nonlinear or non-Gaussian, other variants of the Kalman filter, such as the EnKF, are used to obtain close-to-optimal solutions.” (line 103) needs a reference.
  • The sentence “In such cases, sequential Monte Carlo methods show superior performance over EKF.” (line 153) needs a reference.
  • “… noise is Gaussian distributed” (line 190). How close is a real noise to the Gaussian distribution?
  • Figure 1. It is a disputable idea to place extensive textual description right on a figure. It is better to place it after the figure in the main text.
  • “We added to the online databank realistic motion artifacts, whose generation process is described in our previous paper [32], to mimic real-life scenarios. In this experiment, the ECG data were recorded from a healthy subject during different types of activities.” (line 221). It seems that these two sentences contradict each other. It is not clear from this portion of the text whether the authors added artificial artifacts (“We added to the online databank realistic motion artifacts”) or the authors recorded ECGs with artifacts (“ECG data were recorded from a healthy subject…”)?
  • Some additional numerical parameters of the testing data should be given in “2.4.1. Modified Signals…” Total numbers of records etc. Did the authors use the whole PhysioNet 2013 Challenge databank (with added artifacts) or part of it, or did they add some new signals?
  • “We added to the online databank realistic motion artifacts, whose generation process is described in our previous paper [32].” (line 221). These artifacts are poorly described in paper [32]. It is better to give a more detailed description. What particular types of artifacts were added? What are their realistic amplitudes?
  • Some additional numerical parameters of testing data should be given in “2.4.1. Our Human Data” too. Total numbers of records etc. It is better to focus on the dataset, not on the device in this section. Some data are provided further (line 339), but it is better to describe the database in the corresponding section.
  • Subsections: “2.4.1. Challenge databank”, “2.4.1. Modified Signals…”, “2.4.1. Our Human Data”. The numbers are incorrect.
  • If the time consumption is used as a comparison criterion it should be clearly stated (for example in “2.5.2. Running time” (line 250)). Otherwise it is not clear why the specification of the computer on which the computation was performed is important. “The execution time of each algorithm is calculated from the start of the pre-processing to the end of the fECG R peak detection” (line 253) - no information regarding time parameters is provided further.
  • “The proposed EnKF-based algorithm was run with an ensemble size of N = 70 for single-channel signals from the aforementioned datasets” (line 255). Some reasons or recommendations for choosing N value should be provided.
  • “Figure 2 shows plots of the ECG signals … ”. It is better to specify the Id of the particular signal.
  • “Figure 4 depicts the fECG extraction results from the modified PhysioNet data with added motion artifacts. As seen, the fECG extracted by the EKF was incorrect and its F1 score was reduced significantly (F1=78, see Table 1)” (line 289). This description may lead to the wrong conclusion that Table 1 is based on the data from Figure 4 (several cardiac cycles of the single ECG record). However Table 1 represents the resulting data for the whole dataset (If I understand correctly).
  • In addition to the previous issue: in such investigations raw resulting data are often shown (number of QRS in all records, number of recognized QRS etc.).
  • It is better to add the results for “2.4.1. Our Human Data” to the table too.
  • “In our simulations, we have observed that increasing the size of the ensemble by more than 70 does not improve the performance of the algorithm significantly” This statement can be illustrated by the graph (size of the ensemble and the performance).
  • The sentences “The PhysioNet 2013 Challenge databank used in this work consists of 75 recordings” (line 205) and “…for all the 68 aECG records obtained from the PhysioNet database” (line 326)” (and line 340 too). The numbers are different.
  • “We also plan to correlate with other physiological signals using machine learning and cloud computing [45]” (line 387). I see no cloud computing in [45].

Author Response

(The authors gave the same response as above.)
